# Design and Fabrication of High Performance InGaAs near Infrared Photodetector

**DOI:** 10.3390/nano13212895

**Published:** 2023-11-01

**Authors:** Hezhuang Liu, Jingyi Wang, Daqian Guo, Kai Shen, Baile Chen, Jiang Wu

**Affiliations:** 1Institute of Fundamental and Frontier Sciences, University of Electronic Science and Technology of China, Chengdu 610054, Chinajiangwu@uestc.edu.cn (J.W.); 2School of Information Science and Technology, ShanghaiTech University, Shanghai 201210, Chinachenbl@shanghaitech.edu.cn (B.C.)

**Keywords:** quantum efficiency, photodiode, near infrared, dark current, low capacitance

## Abstract

InGaAs photodiodes have a wide range of important applications; for example, NIR imaging, fiber optical communication, and spectroscopy. In this paper, we studied InGaAs photodiodes with different doping concentration absorber layers. The simulated results suggested that, by reducing the absorber doping concentration from 1 × 10^16^ to 1 × 10^15^ cm^−3^, the maximum quantum efficiency of the devices can rise by 1.2%, to 58%. The simulation also showed that, by increasing the doping concentration of the absorber layer within a certain range, the dark current of the device can be slightly reduced. A PIN structure was grown and fabricated, and CV measurements suggested a low doping concentration of about 1.2 × 10^15^ cm^−3^. Although the thermal activation energy of the dark current suggested a distinct component of shunt dark current at a high temperature range, a dark current of ~6 × 10^−4^ A/cm^2^ (−0.5 V) was measured at room temperature. The peak quantum efficiency of the InGaAs device was characterized as 54.7% without antireflection coating and 80.2% with antireflection coating.

## 1. Introduction

A rich variety of infrared photodetectors have received progressively greater attention over the past few decades, due to increasing demands in both civilian and defense sectors [1,2,3,4,5]. In particular, near infrared (NIR) photodetection, in the spectral region of 0.9–1.7 μm, consistently fulfils research and industry interest in applications for environmental monitoring [6,7], component analysis, and target discrimination [8], where visible and thermal detections are less competent, thanks to abundant information embedded in the NIR optical window. To answer the demand for NIR sensing in wider applications, i.e., food testing/analysis, recycling sorting, etc., high performance photodetectors that can operate at room temperature are highly desired. Ternary III–V compound InGaAs semiconductors and binary II–VI PbS colloidal quantum dots (CQDs) have been widely investigated and developed for large scale focal plane arrays (FPAs), since both can work at room temperature without the necessity of an external cooling apparatus [9,10,11,12]. While the preparation technology for CQDs dramatically simplifies the fabrication of FPAs, particularly their integration with silicon-based readout circuits (ROICs), their performance is yet to be improved in terms of quantum efficiency, bandwidth, etc. [13]. In contrast, the advanced InGaAs detector technology presents additional advantages, including lower power consumption and the ability to achieve high frame rates in imaging, thereby establishing its superiority over CQD NIR detector technology. In this context, an InP lattice-matched InGaAs NIR photodetector proves to be the most favorable choice. In_0.53_Ga_0.47_As has been proved to match well with InP substrate, with a direct bandgap about 0.75 eV, which ensures efficient photon absorption within a wavelength range of 0.9–1.7 µm. Doping strategies have been thoroughly studied for the optimization of InGaAs processing, to reach high material quality and achieve a high device performance [14,15,16,17]. By using a highly doped p-type absorber, Huapu et al. reported InGaAs photodiodes with excellent frequency behavior of 47.5 dBm at 20 GHz [16]. To improve carrier transport in high responsivity and high speed InGaAs photodetectors, graded doping is applied in the absorber layer [17]. Generally, InGaAs PIN photodiodes require very light doping levels (10^15–^10^16^ cm^−3^) of the absorber layer to achieve an optimum performance [18]. With the development of imaging in the near infrared region, the fast-growing demand for large planar arrays calls for a more comprehensive understanding of InGaAs photodetector performance.

In spite of the maturity of InGaAs photodiodes over the past few decades, there has been a resurgence of interest in high-performance InGaAs NIR detectors, specifically those exhibiting a low dark current, low capacitance, and a high quantum efficiency, driven by the increasing demand in emerging applications, notably autonomous driving [19,20,21]. Nonetheless, a dilemma arises when attempting to balance these aforementioned desirable characteristics. For example, large area and low capacitance InGaAs photodiodes can be realized by depleting a thick absorber layer under substantial reverse bias. However, it is challenging to achieve a sharp diffusion profile and a high-quality interface, while the hetero-epitaxial growth of a sufficiently thick InGaAs layer on InP presents a challenge, due to a potential small lattice mismatch and the resultant residual strain. In addition, it is worth noting that a high bias voltage not only introduces trap-assisted tunneling dark current but also poses challenges for focal plane arrays (FPAs). Another challenge arises from the trade-off between a short transient time, which favors a shorter absorber, and a high quantum efficiency, which necessitates a thicker absorber. Previous studies have demonstrated that reducing the carrier concentration within the InGaAs absorption layer can effectively decrease device capacitance when operating at low bias voltages [22]. Nevertheless, although most of these studies concentrated on achieving a low capacitance and thus, high bandwidth, it is important to explore other figures of merit for InGaAs photodiodes featuring an absorption layer with low doping, specifically on dark current and quantum efficiency.

In this work, we investigated the influence of the absorber doping concentration on the performance of an InGaAs photodiode. We then fabricated In_0.53_Ga_0.47_As photodetectors with quasi-intrinsic absorbers and the performance of these devices were analyzed, which may potentially play a guiding role in further research on high-performance InGaAs NIR detectors.

## 2. Methods

The simulations were performed by the commercial software Crosslight APSYS, Version 2021 [23]. The radiative recombination, Auger recombination, Shockley–Read–Hall (SRH) recombination, band-to-band tunneling, and trap-assisted tunneling processes were considered to calculate the dark current of the devices. Figure 1a shows the layer structure of the In_0.53_Ga_0.47_As detector, lattice-matched grown on an InP(100) substrate. The simulations studied four devices with absorber doping concentrations of 1 × 10^15^, 2 × 10^15^, 5 × 10^15^, and 1 × 10^16^ cm^−3^, respectively. A wafer with the exact structure as that in Figure 1a was grown through the metalorganic chemical vapor deposition (MOCVD) technique, except that the absorber was unintentionally doped, to target a low doping concentration below 2 × 10^15^ cm^−3^. The lattice-matched In_0.53_Ga_0.47_As layers were grown on the n^+^-type InP(100) substrate. An n-type 250 nm In_0.53_Ga_0.47_As layer, doped to 2 × 10^18^ cm^−3^, was deposited as an etch stop, followed by a 2.5 μm In_0.53_Ga_0.47_As absorber layer sandwiched by heavily doped InP contact layers. The top p^+^-type InP contact layer is succeeded by a p^++^-type InGaAs layer for improved Ohmic contact.

Following the epitaxial growth process, the wafer was fabricated into circular photodetectors with a diameter of 130 μm. The detector mesas were fabricated using standard photolithography and dry etching techniques. Subsequently, the anti-reflectance (AR) layer was deposited using inductively coupled plasma chemical vapor deposition (PECVD) with an Oxford Plasma pro 100 PECVD 180 system. The thickness of the deposited SiN_x_ AR film was precisely adjusted to 194 nm, which corresponded to achieving destructive interference in SiN_x_ at a wavelength of 1550 nm. The Ti/Pt/Au contact metal was deposited onto heavily doped p+ and n+ contact layers using the DENTON EXPLORER-14 multi-source furnace electron beam evaporation system. The layer thicknesses of the metals were 20, 60, and 120 nm for Ti, Pt, and Au, respectively. A 60s thermal annealing process at 360 °C was employed to facilitate the formation of Ohmic contacts.

## 3. Results and Discussion

Figure 1b shows the simulated dark current density–voltage (J–V) characteristics of the detectors with absorbers of different doping concentrations. The dark current density of the devices increased gradually with a reduction in absorber doping. Specifically, the dark current demonstrated a clear decrease from 2.7 × 10^−5^ A/cm^2^ (−0.5 V), at the lower doping concentration of 1 × 10^15^ cm^−3^, to 3.4 × 10^−6^ A/cm^2^ (−0.5 V) at the higher concentration of 1 × 10^16^ cm^−3^. While the variation in doping concentration may be relatively small, its impact on the dark currents at 300 K was not negligible, the optimization and fine control of the doping concentration may be necessary to achieve the desired device characteristics. The quantum efficiency of all devices was calculated and plotted in Figure 1c. The optical absorption data, which were used for the numerical calculations, was obtained from Ref. [24], and no anti-reflective coating was utilized. It is evident that variations in the absorber doping concentrations resulted in some differences in the quantum efficiency of the detectors. As shown in Figure 1c, the calculated peak quantum efficiency of an unintentionally doped device was 57.9%, which decreased to 56.8% when the absorber doping concentration increased to 1 × 10^16^ cm^−3^.

The variations in dark current and quantum efficiency of the devices with different doping concentrations can be simply explained by the changes in depletion width and the differences in carrier drift velocity. In our case, the broadening of depletion with reducing doping concentration of the absorber layers can be well-explained by the one-side abrupt junction equation, *W* = [2*ε_s_*(*V_bi_* − *V*)/*qN_B_*]^1/2^, where *ε_s_, V_bi_, V, q*, and *N_B_* are the dielectric constant, built-in voltage, external bias voltage, electron charge, and absorber layer doping concentration, respectively. A reduced doping concentration in the low doping side can lead to an increased depletion region width, *W* [25]. As all the doping concentrations were relatively low, the changes in non-radiative recombination rates, such as the Auger recombination and SRH recombination, were marginal. Therefore, the change in depletion width, influenced by the absorber doping concentration, governed the changes in detector performance. As the doping concentration in the absorber layer decreased, the depletion width tended to increase. A broader depletion region implied a greater absorption of incident photons within it, leading to higher quantum efficiency. A wider depletion region also caused a larger diffusion current, which was inversely proportional to the doping concentration of the absorption layer and resulted in a larger generation-recombination current. In addition to the changes in the width of the depletion region, the external voltage that was exerted on the absorption region increased with a decreasing doping concentration of the absorber. Carriers at the neutral region of the absorption layers were more easily able to reach the contact terminals of the detector. Combining the broadening of the depletion region width and the increase in drift velocity, both the dark current and the quantum efficiency dropped within a certain range as the doping concentration increased.

The J–V curve of the fabricated device was measured at 300 K and plotted with the simulated result of the device with an absorber doping concentration of 1 × 10^15^ cm^−3^, as shown in Figure 2a. The simulation result was more than one order of magnitude lower, compared with the experimental measurements of the dark current. This inconsistency may be attributed to the presence of a significant shunt current, which was not considered in the simulation. To gain further insights into the dark current mechanism, the dark current of the fabricated device was measured from 77 K to 300 K, as shown in Figure 2b. At low temperatures, the dark current was too low for the measurement system to collect reliable dark current data when the bias voltage is relatively small. However, tunneling currents, e.g., the band-to-band tunneling current, were activated when the bias voltage was beyond a certain range. At the low temperature of 77 K, the dark current density was only about 5.9 × 10^−8^ A/cm^2^, even under a large reverse bias of up to 5 V. With an increasing temperature, the dark current density showed a distinct increase. Arrhenius plots of the device dark current at a reverse bias of 0.5 V are presented in Figure 2c. At low reverse bias voltages, the temperature-dependent dark current can be described by an effective thermal activation energy, given by *I_d_~exp(−E_a_/kT)*, where *E_a_*, *k*, and *T* represent the activation energy, Boltzmann’s constant, and the temperature, respectively [26]. The extracted thermal activation energy serves as a valuable tool for assessing the primary dark current mechanism across various temperature ranges. The activation energy of the device was calculated to be 0.343 eV, which was much lower than the measured bandgap energy (0.708 eV) of In_0.53_Ga_0.47_As. This result suggests that the main dark current component was not governed by diffusion, but rather by the significant generation-recombination or shunt mechanism at high temperature. The activation energy *E_a_* for the diffusion current (*I_diff_*), generation-recombination current (*I_gr_*), shunt current (*I_sh_*), and tunneling current (*I_tun_*), were theoretically determined to be *E_g_*, *E_g_/*2**, *E_g_/*2**, and *E_g_/*4**, respectively [27]. With an activation energy of 0.343 eV, which closely approaches *E_g_/*2** in the temperature range of 200–300 K, the dominant mechanisms governing the dark current of the detector were the generation-recombination process or shunt current. However, due to the relatively low doping concentration and the lattice matching of the In_0.53_Ga_0.47_As epilayers to the substrate, the dominant factor above 200 K was ascribed to the shunt current. This observation aligned with the higher measured dark current, compared to the simulation results. Based on the above information and analysis, it is evident that the shunt current demands our immediate attention, particularly in the case of smaller photodiodes. Minor factors, such as tunneling current and generation-recombination current, may also contribute to the dark current. For imaging applications, dark current is an important factor, especially when the light power is low. Typically reported InGaAs arrays, where the area-dependent dark current is not neglected, have a dark current density around 1 nA/cm^2^ at room temperature [28]. From these perspectives, further work should be made on better surface passivation.

The performance of the diodes was measured through capacitance–voltage (C–V) measurements at room temperature. In Figure 3, the experimental C–V curves for diodes of varying diameters are presented. As illustrated in Figure 3a, even at a relatively low voltage of 5 V (typically recommended for regular packaged diodes), the C–V curve flattened, indicating near-complete depletion, with capacitance reaching its minimum value. The measurements for detectors with diameters smaller than 200 μm reveal that the detector capacitance remained below 5 pF at a low voltage of 2 V, suggesting low intrinsic doping within the unintentionally doped InGaAs layers. In Figure 3b, the measured capacitance of detectors with different diameters, measured at 2 V and 4 V, is depicted as a function of device diameter. Notably, a parasitic capacitance of 114 fF was extracted. For the one-side abrupt junction, i.e., an asymmetrically doped p^+^-n junction, the inverse capacitance squared (1/*C^2^*) was a linear function of the applied reverse biased voltage, and the slope of the curve was inversely proportional to the doping concentration of the unintentionally doped InGaAs absorber region [25]. Therefore, the doping concentration of the n-type absorber can be estimated using the slope of the 1/*C^2^*–*V* plot and calculated as *N_D_* = $2/\left[q\varepsilon A^{2}d\left(1/C^{2}\right)/dV\right]$, where *q* is the elementary charge, *ε* is the dielectric constant of the absorber layer, *A* is the device area, *C* is the measured capacitance, and *V* is the applied bias [29]. For further analysis, the junction capacitances of the diodes of different diameters were measured under different reverse biases at room temperature, as shown in Figure 3. The plots of 1/*C^2^* and reverse bias voltage are shown in Figure 3c,d. For all devices, the capacitance decreased steadily as the reverse bias increased, while it increased with higher doping concentrations. The doping concentrations of the absorber regions with different detector diameters, which were extrapolated from the curves, fell in the range of 1.20–1.64 × 10^15^ cm^−3^; as shown in Figure 3c,d, the measured doping concentrations were 1.20 × 10^15^ cm^−3^, 1.21 × 10^15^ cm^−3^, 1.30 × 10^15^ cm^−3^, and 1.64 × 10^15^ cm^−3^ for detectors with diameters of 500 μm, 350 μm, 180 μm, and 100 μm, respectively. With a smaller device diameter, the doping concentration was higher, and the reasons behind such a discrepancy were related to the mesa edge effects on the doping measurements by the C–V technique. The measured capacitance can be deviated from its intrinsic value due to fringing effects [30]. Nonetheless, the calculated values from the C–V measurements closely aligned with the target growth value.

The quantum efficiency of the fabricated InGaAs detector was measured in the front-illumination mode at 300 K and is plotted in Figure 4. The quantum efficiency of the photodiode is calculated by *η* = *R* × *hc/qλ*, where *R* is the measured responsivity, *q* is the electron charge, *h* is the Planck constant, *c* is the speed of light, and *λ* is the incident light wavelength. As shown in Figure 4a, the quantum efficiency of the detector, when coated with anti-reflective (AR) layers, exhibited a decent performance across the entire NIR spectrum range up to 1.7 µm, the 50% cutoff wavelength of the photodiode. At zero bias, a peak quantum efficiency of approximately 42% was achieved at 1.4 µm. Considering the optical loss from reflection, the relatively high quantum efficiency demonstrates that a 2.5 μm absorber layer thickness provided sufficient absorption volume. This peak efficiency was saturated at 52.4% with a rather low bias voltage of 0.5 V, implying the effective depletion of the near-intrinsic absorber at a low bias voltage. The quantum efficiency slightly decreased in the short wavelengths, which can be attributed to the higher absorption loss of photons of short wavelengths by the highly doped top contact layer. It is commonly observed in PIN photodiodes that pronounced photo-absorption and carrier recombination in the uppermost p-type layer can lead to a decrease in quantum efficiency in the short wavelength range [31]. However, this decline can be effectively mitigated by etching away the top contact layer. Notably, the measured quantum efficiency aligned excellently with the simulation results, despite discrepancies in dark currents. This agreement further indicates that non-radiative losses, attributed to dislocations, played a minor role in the device’s performance. Furthermore, it is also worth pointing out that high quantum efficiencies at lower bias voltages can be achieved, which is beneficial to reduce power consumption in imaging and sensing systems. The inset of Figure 4a displays the photo-responsivity of the detector measured at room temperature. The detector had a responsivity of 0.48 A/W at the wavelength of 1550 nm under zero bias, and an increased responsivity over 0.6 A/W with a bias voltage over 0.5 V. The responsivity of the photodetector increased with bias due to the wider depletion region and larger electric field in the depletion region of the reverse-biased diode, enhancing the separation and drift of photo-generated carriers. Upon the application of an anti-reflective (AR) coating, the responsivity was boosted to 0.72 A/W at the wavelength of 1550 nm under zero bias, and nearly 1.0 A/W with a bias voltage over 0.5 V. Therefore, the peak quantum efficiency experienced a notable enhancement, reaching 60% at zero bias voltage and exceeding 80% under bias voltages of >0.5 V. Our results are closely aligned with the reported InGaAs PDs for imaging applications, indicating the good material quality and device design [32]. The number of photons absorbed relies on the quantity of light coupled into photodiodes and transmitted through the InGaAs absorption region. Hence, the noteworthy quantum efficiency can be ascribed to both the anti-reflective coating and effective optical conversion in the device. Further optimization of the absorber thickness and passivation holds the potential to yield an even higher performance.

## 4. Conclusions

In this study, we studied the impact of light doping within the absorber layer on the performance characteristics of near-infrared (NIR) InGaAs photodiodes. The utilization of low doping levels offers advantages in terms of reduced capacitance and enhanced quantum efficiency, albeit at the cost of a slightly elevated dark current. Future efforts will aim to further reduce the shunt current by improving surface passivation. Specifically, we fabricated In_0.53_Ga_0.47_As photodetectors featuring quasi-intrinsic absorbers, resulting in measured doping concentrations of approximately 1.2 × 10^15^ cm^−3^. These detectors exhibit a low capacitance when operated at a low voltage of −2 V, which is favorable for a high frame rate imager. Additionally, the application of an anti-reflective (AR) coating enabled the attainment of high quantum efficiency levels, reaching up to 80%. The work may be helpful for facilitating further reductions in the size, weight, and power consumption of InGaAs photodiodes, thereby facilitating a broader range of imaging and sensing applications in the near infrared range.

## Figures and Tables

**Figure 1 nanomaterials-13-02895-f001:**
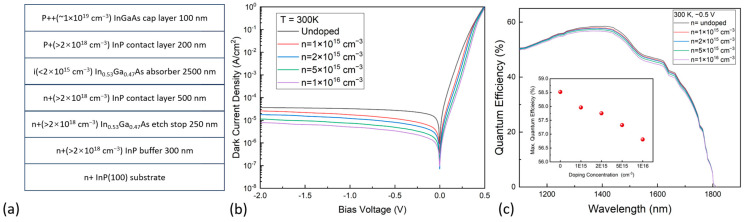
(**a**) Layer structure of an InGaAs photodiode. (**b**) The simulated dark current density–voltage characteristics of the InGaAs photodiodes with different doping concentrations in the absorber. (**c**) Simulated quantum efficiency spectra of the devices with different absorber doping concentrations. Insert in (**c**) shows the maximum quantum efficiencies with different doping concentrations.

**Figure 2 nanomaterials-13-02895-f002:**
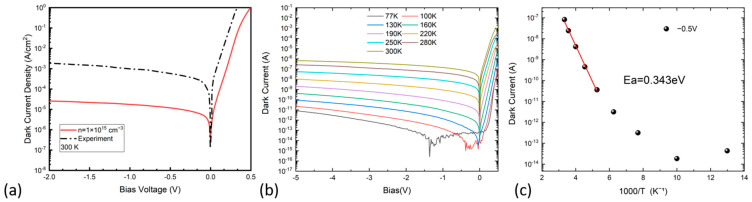
(**a**) Simulated and measured J–V curves of the InGaAs photodiode with a low doping concentration of ~1.0 × 10^15^ cm^−3^. (**b**) Temperature dependent I–V curves measured from the fabricated InGaAs photodiode. (**c**) Arrhenius plot of the dark current of the detector measured with a bias voltage of −0.5 V.

**Figure 3 nanomaterials-13-02895-f003:**
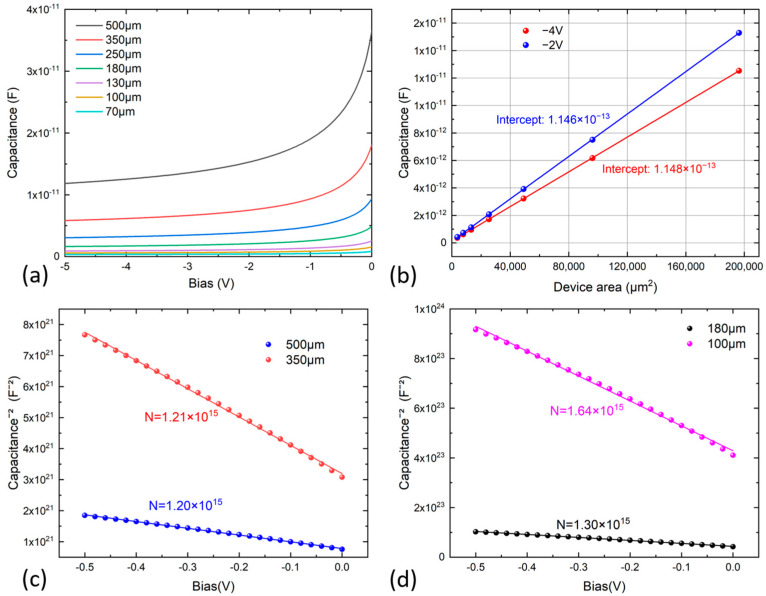
(**a**) C–V curves of the fabricated InGaAs photodiodes with different diameters. (**b**) Plot of capacitances of photodiodes as a function of device area. (**c**) 1/*C^2^* vs. reverse voltage plot for the photodiodes with mesa diameters of 500 μm and 350 μm at 300 K. (**d**) 1/*C^2^* vs. reverse voltage plot for the photodiodes with mesa diameters of 180 μm and 100 μm at 300 K.

**Figure 4 nanomaterials-13-02895-f004:**
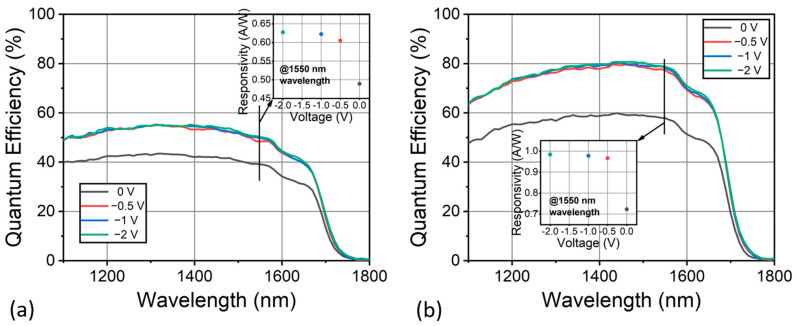
(**a**) Quantum efficiency of the fabricated InGaAs photodiode without AR coating under different bias voltages at 300 K. (**b**) Quantum efficiency of the fabricated InGaAs photodiode with AR coating under different bias voltages at 300 K. The insets show the measured photoresponsivity at 1500 nm as a function of bias voltages.

## Data Availability

The data presented in this study are available on request from the corresponding author.

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
