# Peer review of "Design and Fabrication of High Performance InGaAs near Infrared Photodetector"

_nanomaterials, 2023, doi:10.3390/nano13212895_

Round 1
Reviewer 1 Report
Comments and Suggestions for Authors
REVIEW
on the manuscript “Design and fabrication of high performance InGaAs near infra-red photodetector”
by Hezhuang Liu, Jingyi Wang, Daqian Guo, Shen Kai, Baile Chen, Jiang Wu
In the presented manuscript the results of experimental and model investigations of the photodetectors of near-infrared range based on InGaAs/InP multilayered structures are presented. Dependences of pin photodiode characteristics on the doping concentration in the absorber layer are calculated and measured. Comparison with photosensitive structure with undoped absorber is carried out. Dark current-voltage characteristics, capacitance-voltage characteristics and quantum efficiency of the proposed photodetectors are analyzed. The influence of doping concentration and device area on the parameters of photodetector is studied. These structures and technological approach to the device design should be useful for further device applications in optoelectronics.
Chosen methods are adequate and up-to-date, while results of experimental research are reliable. Overall, the paper is organized and written well and at high scientific level.
However, in the reviewer’s opinion, the manuscript should be improved before publication. The details are listed below.
1. Abstract, The sentence “The simulated results suggested that undoped absorber and doping concentrations of 1×1015 and 2×1015 cm-3 support a high quantum efficiency of around 60%.” is confusing. Please, rewrite.
2. Figure 1b: “Bios voltage” should be corrected to “Bias voltage” in the axis name.
3. Figure 1c: Inset may be added, showing the values of maxima of quantum efficiency in details.
4. Methods section: Anti-reflection layers should be described.
5. Methods section: Ohmic contacts should be specified.
6. Figure 2c: How authors can describe the behavior of the Arrhenius plot where it is non-linear (at low temperatures).
7. Figure 4 and Figure 1: Quantum efficiency units may be done consistent.
8. Why CV-characteristics under light illumination are not presented?
9. Comparison of the obtained values of photodetector parameters with some other photodetectors from the literature should be carried out.
10. Additionally, the language of manuscript should be double-checked and some typos should be removed, for example “driven0020by” in Introduction, “can leads” and “1the short” in Results and Discussion section.
Conclusion: The presented manuscript may be published in the Nanomaterials journal after moderate revision.
Comments on the Quality of English LanguageThe language of manuscript should be double-checked and some typos should be removed.
Reviewer 2 Report
Comments and Suggestions for Authors
I have comments on the paper :
InGaAs photodiodes on InP substrate having cut-off wavelength at around 1.6-1.7µm are commercially available (see InGaAs devices from Teledyne/Judson Technology for example).
Innovation for InGaAs photodetector would be to extend the cut-off wavelength to reach the SWIR domain (lambda higher than 2.2µm) or to develop new high performance structure on Si substrate for integrated photonic systems.
- In the introduction, the authors should better justify the objectives of this study.
- error in the abstract concerning peak QE at 54% without and 80% with ARC
- why are the calculated dark current values reported at -500 mV? Was the
calculated quantum efficiency achieved at this voltage?
- At room temperature, the photodiode current in dark condition should be diffusion limited. How can we explain that an activation energy corresponding to a GR current is extracted from the Arrhenius plot ?
Round 2
Reviewer 1 Report
Comments and Suggestions for Authors
REVIEW
on the manuscript “Design and fabrication of high performance InGaAs near infra-red photodetector”
by Hezhuang Liu, Jingyi Wang, Daqian Guo, Shen Kai, Baile Chen, Jiang Wu
In the presented manuscript the results of experimental and model investigations of the photodetectors of near-infrared range based on InGaAs/InP multilayered structures are presented. Dependences of pin photodiode characteristics on the doping concentration in the absorber layer are calculated and measured. Comparison with photosensitive structure with undoped absorber is carried out. Dark current-voltage characteristics, capacitance-voltage characteristics and quantum efficiency of the proposed photodetectors are analyzed. The influence of doping concentration and device area on the parameters of photodetector is studied. These structures and technological approach to the device design should be useful for further device applications in optoelectronics.
Chosen methods are adequate and up-to-date, while results of experimental research are reliable. Overall, the paper is organized and written well and at high scientific level.
The revision of the manuscript is satisfactory. The authors have answered all the questions and made necessary amendments.
Conclusion: The presented manuscript may be published in the Nanomaterials journal.
Reviewer 2 Report
Comments and Suggestions for Authors
In the introduction, comments were added but to better justify the objectives of this study, it is necessary to summarize the results currently known on InGaAs photodiodes in terms of dark current, quantum efficieny and doping and how the paper would be important for the next FPA generation .
Round 3
Reviewer 2 Report
Comments and Suggestions for Authors
Authors have taken into account remarks made by the reviewers
Comments on the Quality of English Languageno error identified